# A Bayesian approach to reveal the key role of mask wearing in modulating projected interpersonal distance during the first COVID-19 outbreak

Matteo P. Lisi[1,2]*, Marina Scattolin[1,2], Martina Fusaro[2], Salvatore Maria Aglioti[1,2]*

**1** Sapienza University of Rome and Center for Life Nano- & Neuro-Science, Fondazione Istituto Italiano di Tecnologia (IIT), Rome, Italy, **2** Social Neuroscience Laboratory, Fondazione Santa Lucia, Rome, Italy

\* matteo.lisi@uniroma1.it (MPL); salvatoremaria.aglioti@uniroma1.it (SMA)

**Data Availability Statement:** All the data and codes for the analysis are available from the Mendeley repository: https://data.mendeley.com/

## Abstract

Humans typically create and maintain social bonds through interactions that occur at close social distances. The interpersonal distance of at least 1 m recommended as a relevant measure for COVID-19 contagion containment requires a significant change in everyday behavior. In a web-based experimental study conducted during the first pandemic wave (mid-April 2020), we asked 242 participants to regulate their preferred distance towards confederates who did or did not wear protective masks and gloves and whose COVID-19 test results were positive, negative, or unknown. Information concerning dispositional factors (perceived vulnerability to disease, moral attitudes, and prosocial tendencies) and situational factors (perceived severity of the situation in the country, frequency of physical and virtual social contacts, and attitudes toward quarantine) that may modulate compliance with safety prescriptions was also acquired. A Bayesian analysis approach was adopted. Individual differences did not modulate interpersonal distance. We found strong evidence in favor of a reduction of interpersonal distance towards individuals wearing protective equipment and who tested negative to COVID-19. Importantly, shorter interpersonal distances were maintained towards confederates wearing protective gear, even when their COVID-19 test result was unknown or positive. This protective equipment-related regulation of interpersonal distance may reflect an underestimation of perceived vulnerability to infection; this perception must be discouraged when pursuing individual and collective health-safety measures.

## Introduction

On March 11, 2020, the World Health Organization described the COVID-19 outbreak as a pandemic to signal that the new coronavirus disease had spread across continents, covering large parts of the world. The severe acute respiratory syndrome coronavirus-2 (SARS-CoV-2), which is the virus responsible for the emergence of the COVID-19 disease, was found to be

datasets/jw3sbz2nkv/1 (DOI: 10.17632/jw3sbz2nkv.1).

**Funding:** This work was supported by a European Research Council (ERC, https://erc.europa.eu/) Advanced Grant 2017, Embodied Honesty in real world and digital interactions (eHONESTY) to SMA (grant number 789058), Avvio alla Ricerca (2019) awarded by La Sapienza University of Rome (https://www.uniroma1.it/en) to MF (grant number AR21916B88690F78), Avvio alla Ricerca (2020) awarded by La Sapienza University of Rome to MPL (grant number AR120172B17A70D1). The funders had no role in study design, data collection and analysis, decision to publish, or preparation of the manuscript.

**Competing interests:** The authors have declared that no competing interests exist.

transmissible during social interactions, when particles emitted from an infected person's respiratory system may enter another's [1]. To limit gatherings and close-range interactions, multiple governments imposed the closure of many public places. These closures and other measures of transmission containment, such as handwashing and use of face masks [2], have been widely adopted in conjunction with maintaining interpersonal distances of at least 1 m [3]. The need to regulate the minimum distance during in-person interactions is justified by the observation that, although humans tend to keep themselves at about 1 m from unfamiliar individuals [4], this distance reduces when interacting with acquaintances and friends [5]. Crucially this pattern seems to hold across different countries [5], suggesting that the imposed governmental measures sought to change a globally established, everyday behavior. While the reasons behind enforcement of such distancing rules are clear, it remains to be clarified which dispositional and situational factors may impact adherence to interpersonal distancing measures. By providing insights on this topic, social and behavioral sciences can support human responses to pandemics [6, 7].

Research in proxemics, the study of interpersonal spatial behavior [8, 9], has defined interpersonal distance (IPD) as the separation zone that individuals keep between themselves and others [10]. IPD is shaped by situational factors such as social threat [11, 12] and interpersonal attraction [13], as well as individual characteristics such as morality [14] and prejudice [15]. Ultimately, the appropriate IPD appears to be automatically regulated according to distance-related feelings of personal comfort [16]. Although previous research has largely investigated IPD under regular circumstances, much less is known about the factors influencing the regulation of IPD during the spread of infectious diseases. A limited body of research suggests that greater distances are kept when others are improperly perceived as contagious (i.e., people with AIDS) or as a threat to an individual's health [17, 18]. Moreover, two studies conducted during the first COVID-19 outbreak [19, 20] showed that a smaller IPD was preferred when the other person was wearing a mask. One possible interpretation is that the sight of a person wearing a face mask triggers a feeling of safety. It is important to note that, if not accompanied by the appropriate IPD, wearing a mask is not in itself sufficient to prevent contagion [21]. Therefore, one crucial aspect to clarify is whether wearing a mask can reduce the IPD even when the other person is contagious. If this is the case, the erroneous belief that the use of protective equipment is enough to prevent contagion may have potentially dangerous effects.

It is worth noting that modulations of IPD during a pandemic may not only mirror self-protective motives, but also affiliative [22] and cooperative ones [23]. In fact, previous research showed that prosocial individuals are more likely to follow physical distancing [24, 25], while prosocial messages appear to foster compliance with health behaviors [26].

Understanding which of these variables are more influential on IPD behavior is crucial in the current global context, where policies that effectively reduce contagion are fundamental.

Given the circumstances preventing in-person testing, we used the Interpersonal Visual Analogue Scale (IVAS), a validated, self-report measure of IPD [27]. In the IVAS, participants were presented with a silhouette and an avatar's profile on a computer screen. The silhouette represented the participant and the avatar represented a possible unknown individual. Participants were asked to indicate the shortest distance between themselves and the other person that they would feel comfortable maintaining. Both male and female interactants were considered, and distance was indicated along a horizontal line. Since the aim of this study was to investigate whether being at risk of infection modulates participants' predicted IPD, the avatar representing the other person was associated with a negative, positive or unknown COVID-19 test result. In addition, the avatar could be wearing protective equipment (i.e., mask and gloves) or not. Factors hypothesized to play a role included (a) the perceived vulnerability to a disease, and (b) the perceived severity of the situation in the country, which were relevant for

evaluating attitudes towards the threat itself. We also explored (c) the role of individual differences in levels of physical and virtual contact prior to participation. Finally, we aimed to assess the possible role of different styles of (d) moral thinking (individualization-oriented and binding-oriented), (e) attitudes toward quarantine, and (f) altruism.

We expected participants to maintain a greater distance when others were not wearing protective equipment and when they were identified as positive to COVID-19. The shortest distance was expected to be observed when the other person was wearing a mask and gloves and had received a negative diagnosis of COVID-19. We included a condition in which COVID-19 test results were unknown. This was our control condition and was used to estimate (i) how participants may react to strangers approaching them in everyday situations, and (ii) how reactions may change when the other person is wearing protective equipment or not. We expected higher levels of perceived severity of the situation in the country of participants and their perceived vulnerability to a disease to be associated with greater IPD. As to the affiliative domain, we expected our results to show one of the following two paths: on the one hand, participants who engaged in fewer virtual and physical contact may display a stronger tendency to distance themselves from others [28] compared to participants with more frequent contacts. On the other hand, and in accordance with the contact hunger hypothesis [22], the opposite pattern of results could appear: people who had engaged in more frequent and recent social contacts at the time of testing may not feel the need for closeness that people who had engaged in fewer and less frequent contacts may feel. We expected that participants' positive attitudes toward quarantine may be associated with greater IPD across all conditions. In addition, since binding (vs. individualizing) moral intuitions are more strongly correlated with dispositional germ aversion [29], we expected binding moral thinking styles to contribute to the tendency to maintain a greater IPD. Lastly, if prosocial motives play a role, higher levels of altruism should predict greater distance. In order to avoid overfitting and to select only the relevant variables, we used a model selection approach [30].

## Materials and method

### Participants

All procedures were approved by the Ethics Committee of the Department of Psychology, University of Rome "La Sapienza" (Prot. n. 0000612) and in accordance with the ethical standards laid down in the Declaration of Helsinki (2013).

A power analysis using MorePower [31] indicated that a sample size of 238 participants was necessary to detect a small effect size ($\eta2 = 0.02$) with a power of 0.80. This analysis was performed for a repeated measure design 2 (Participant's Gender: Male/Female) x 2 (Other Avatar's Gender: Male/Female) x 2 (Protective Equipment: Worn/NotWorn) x 3 (COVID-19 Test Result: Positive/Negative/Unknown).

Participants were recruited through the online platform Prolific [32] and were compensated with $2.28 USD ($6.50 USD per hour) for their participation. Written informed consent was obtained from all participants. Data collection started on April 16, 2020, and ended on April 22, 2020.

Of the original 250 participants, eight were excluded due to failure in two or more attentional checks. A total of 242 international participants (100 women) were included in the final sample. Demographic characteristics of the sample, as well as a list of all countries of residence, are presented in Table 1. All countries of residence involved in the data collection had an average Government Stringency Index (a composite measure of the strictness of contagion policies in each country, based on nine response indicators, ranging from 0 to 100) [33] greater than 60 (see Table 1).

**Table 1. Demographic characteristics for each of the 28 countries of residence included in the study.**

| Country of residence | Gender | | | Age Range | | | Government Stringency Index |
|---|---|---|---|---|---|---|---|
| | Total | Male | Female | 18–34 | 35–55 | 55 or more | Mean (between 16–22 April 2020) |
| Australia | 4 | 4 | 0 | 3 | 1 | 0 | 70.5 |
| Austria | 3 | 1 | 2 | 3 | 0 | 0 | 77.8 |
| Belgium | 2 | 1 | 1 | 2 | 0 | 0 | 81.5 |
| Canada | 3 | 3 | 0 | 3 | 0 | 0 | 72.7 |
| Czech Republic | 4 | 4 | 0 | 4 | 0 | 0 | 67 |
| Denmark | 1 | 0 | 1 | 1 | 0 | 0 | 68.5 |
| Estonia | 8 | 2 | 6 | 7 | 1 | 0 | 77.8 |
| Finland | 2 | 1 | 1 | 2 | 0 | 0 | 61.8 |
| France | 7 | 5 | 2 | 6 | 1 | 0 | 87.9 |
| Germany | 4 | 3 | 1 | 4 | 0 | 0 | 76.8 |
| Greece | 16 | 12 | 4 | 13 | 3 | 0 | 84.2 |
| Hungary | 9 | 5 | 4 | 8 | 1 | 0 | 76.8 |
| Ireland | 3 | 2 | 1 | 3 | 0 | 0 | 90.7 |
| Israel | 3 | 3 | 0 | 3 | 0 | 0 | 89.9 |
| Italy | 26 | 15 | 11 | 24 | 2 | 0 | 93.5 |
| Latvia | 2 | 0 | 2 | 2 | 0 | 0 | 69.4 |
| Mexico | 2 | 2 | 0 | 2 | 0 | 0 | 82.4 |
| Netherlands | 4 | 3 | 1 | 4 | 0 | 0 | 79.6 |
| New Zealand | 3 | 2 | 1 | 3 | 0 | 0 | 96.3 |
| Norway | 2 | 2 | 0 | 2 | 0 | 0 | 76.4 |
| Poland | 34 | 23 | 11 | 31 | 3 | 0 | 83.3 |
| Portugal | 30 | 17 | 13 | 22 | 6 | 2 | 82.4 |
| Slovenia | 5 | 4 | 1 | 4 | 1 | 0 | 88.2 |
| South Africa | 1 | 0 | 1 | 0 | 1 | 0 | 87.9 |
| Spain | 10 | 7 | 3 | 8 | 2 | 0 | 85.1 |
| Sweden | 1 | 1 | 0 | 1 | 0 | 0 | 64.8 |
| UK | 48 | 15 | 33 | 31 | 13 | 4 | 79.6 |
| USA | 5 | 5 | 0 | 4 | 1 | 0 | 72.7 |
| **Total** | **242** | **142** | **100** | **200** | **36** | **6** | |

## Procedure

The experiment was conducted using PsyToolkit [34, 35]. After reading general information concerning the study, participants could check the informed consent page and agree to take part in the study. Only those who gave their consent to participate could begin the survey. Participants always completed the demographic information and the questionnaires before the IVAS task. The silhouette representing participants in the IVAS task was selected based on each participant's gender (task version "Female-Self" and "Male-Self"). The silhouette was pictured in a standing position on a marker at the left end of a line and facing the right end of the line (see Fig 1). The height of participants' silhouettes and of the other person's virtual avatar were matched. The virtual avatars were realized using MakeHuman Community 1.2.0 (http://www.makehumancommunity.org); the pictures of the avatars were taken using Unity v.2019.4.15f1 (https://unity.com). Before beginning the IVAS, participants were provided with the following instructions: "Imagine that you are the person on the left of the line and that you cannot turn nor move. Then, imagine that the other person, depicted on the line, begins

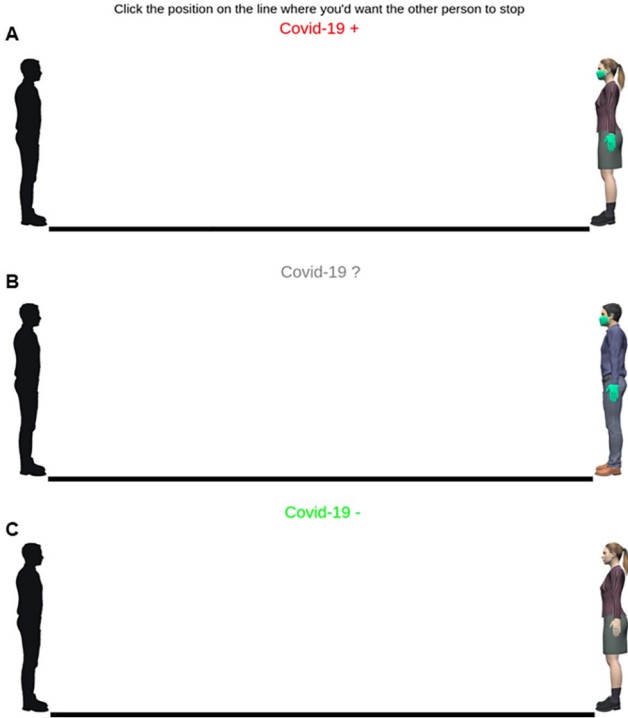

**Fig 1. Example of experimental stimuli.** Participants were instructed to imagine to be the person on the left side, represented by a gender-matched, black silhouette, and to indicate the distance to the other person (female or male, represented in A-C and B, respectively) that they would feel comfortable keeping. The other person could be wearing protective equipment (A-B) or not (C). The label appearing on the upper part of the screen indicated the COVID-19 test result of the other person: a Positive test result was represented by a "+" and displayed in red (A); an Unknown test result was represented by a "?"and displayed in gray (B); a Negative test result was represented by a "-"and shown in green (C).

walking toward you. You should indicate how close you would allow this person to approach you while still being comfortable with that distance. To indicate where the other should stop, click on the horizontal line. Then, press 'Next' to move to the following trial. During the task you will be approached by men and women, that may or may not be wearing masks and gloves. On the top center of the screen you'll read some information about the results of their COVID-19 test: a red sign reporting 'COVID-19 +' indicates a person that tested positive; a green 'COVID-19 –' indicates a person whose results were negative; a grey 'COVID-19?' indicates that the person was not tested or that results are unknown." In each trial, participants were reminded of these instructions by the sentence: "Click the position on the line where you'd want the other person to stop." This reminder was continuously displayed on the top-center of the screen, above the indicator of COVID-19 Test Result (Fig 1).

For each condition (i.e., the combination of factors Other Avatar's Gender x Protective Equipment x COVID-19 Test Result), a total of three trials was presented. Three catch trials were included to ensure that participants were paying attention during the task. In these trials, participants were asked to place the other person's avatar on the far-left end of the line. A total of 39 trials were presented to participants. For each trial, the distance between the participant's silhouette and the other person's avatar was calculated considering that the left end of the horizontal line corresponded to 0 while the right end of the line corresponded to 100.

## Measures

**Perceived Severity of the Situation relative to the COVID-19 outbreak in the country.** Perceived Severity of the Situation concerning the COVID-19 outbreak was assessed through participants' answers to the following question: "In your opinion, how serious is the situation related to COVID-19 in your country?" Participants rated this on a VAS ranging from 0 (labelled as *not serious at all*) to 100 (*extremely serious*).

**Physical and Virtual Contact.** Physical Contact was assessed by means of the question "How often did you have PHYSICAL contacts (for instance, hugs, cuddling, handshakes, etc.) in the last two weeks?" Participants provided their response on a 5-point Likert scale: *never* (1), *rarely* (2), *sometimes* (3), *often* (4), *always* (5). Participants rated frequency of Virtual Contact on a single-item, 5-point Likert scale: *never* (1), *rarely* (2), *sometimes* (3), *often* (4), *always* (5). The question was the following: "How often did you have VIRTUAL contacts (for instance, through Skype, Zoom, WhatsApp, etc.) in the last two weeks?"

**Moral Foundations Questionnaire.** This is a self-report questionnaire that contains 30 items related to harm, fairness, in-group loyalty, respect for authority, and purity (six items for each foundation) [36].

We computed scores for the *individualizing* foundations (mean of the harm/care and fairness/reciprocity subscales; Cronbach's $\alpha$ = 0.77) and the *binding* foundations (mean of the in-group/loyalty, authority/respect, and purity/sanctity subscales; $\alpha$ = 0.85); these values were used in subsequent analyses since we were interested in *individualizing* and *binding* foundations broadly. The focus on the well-being of individuals is observed in association with individualizing approaches to moral thinking [37]. People who rely on this style of moral thinking focus on protection of individuals from harm and unfairness and consider individuals as the center of moral regulations [36]. On the other hand, binding foundations favor moral evaluations that are group-oriented and value authority.

**Perceived Vulnerability to Disease.** Participants completed an adapted version of a questionnaire assessing individual differences in perceived vulnerability to disease [38]. Specifically, we dropped one item ("I avoid using public telephones because of the risk that I may catch something from the previous user") because of its poor relevance to the contemporary context, where the majority of people use cell phones. Consequently, a 14-item version was employed. The overall score (Cronbach's $\alpha$ = 0.75) was used for our analysis.

**Public Attitudes Toward Quarantine.** This is a 15-item questionnaire developed by Tracy and colleagues [39]. Participants were asked to rate each sentence on a 5-point Likert scale ranging from *strongly disagree* to *strongly agree*. The Justification subscale (Cronbach's $\alpha$ = 0.68) was entered in the analysis to investigate the relationship between the agreement with the use of quarantine and the interpersonal distance regulation.

**Self-Report Altruism Scale.** For assessment of participants' altruism, the scale developed by Rushton and colleagues [40] was employed. This includes 18 items measuring helping or altruistic traits based on the frequency of helping behaviors. Cronbach's $\alpha$ was 0.84.

**Sexual Orientation.** For the assessment of participants' sexual orientation, we used the Kinsey Scale [41]. Participants provided their response on a 8-point Likert scale ranging from *exclusively heterosexual* (1), to *exclusively homosexual* (7), and also including *no socio-sexual contacts or reactions* (8).

The full survey can be found in the S1 File.

# Results

## Model comparison

A Bayesian analysis approach was applied. This approach differs from the one used within the standard framework of frequentist null-hypothesis significance testing (NHST) in that it allows evidence to be obtained in favor of the null hypothesis and discrimination between "absence of evidence" and "evidence of absence" [42].

Data analyses were computed using the programming language R [43] by means of the RStudio interface [44]. Preparation and plotting of data were performed using several tidyverse packages [45]; modeling and inference were performed using the brms package [46], which is based on the probabilistic programming language Stan [47]. Packages emmeans [48] and bayestestR [49] were employed for computing contrasts between posterior distributions, credible intervals, and Bayes factors.

The score along the 0–100 Visual Analogue Scale was used as a measure of the distance that participants preferred to keep between themselves and other person's avatars. This measure was set as the outcome of our analysis. We first graphically inspected the univariate and bivariate distributions of outcome and predictor variables. This was done in order to (i) check data

**Table 2. Formulas for each model.**

| Model | Formula |
|---|---|
| Model 0 | *Distance ~ 1 + (1 | Country) + (1 |Participant)* |
| Model 1 | *Distance ~ 1 + Protective Equipment × COVID-19 Test Result + Participant's Gender × Other Avatar's Gender + (1 | Country) + (1 |Participant)* |
| Model 2 | *Distance ~ 1 + Protective Equipment × COVID-19 Test Result + Participant's Gender × Other Avatar's Gender + Perceived Severity of the situation in the Country + (1 | Country) + (1 |Participant)* |
| Model 3 | *Distance ~ 1 + Protective Equipment × COVID-19 Test Result + Participant's Gender × Other Avatar's Gender + Perceived Severity of the situation in the Country + Virtual Contact + (1 | Country) + (1 | Participant)* |
| Model 4 | *Distance ~ 1 + Protective Equipment × COVID-19 Test Result + Participant's Gender × Other Avatar's Gender + Perceived Severity of the situation in the Country + Virtual Contact + Physical Contact + (1 | Country) + (1 |Participant)* |
| Model 5 | *Distance ~ 1 + Protective Equipment × COVID-19 Test Result + Participant's Gender × Other Avatar's Gender + Perceived Severity of the situation in the Country + Virtual Contact + Physical Contact + Perceived Vulnerability to Disease + (1 | Country) + (1 |Participant)* |
| Model 6 | *Distance ~ 1 + Protective Equipment × COVID-19 Test Result + Participant's Gender × Other Avatar's Gender + Perceived Severity of the situation in the Country + Virtual Contact + Physical Contact + Perceived Vulnerability to Disease + Individualizing Moral Foundation + (1 | Country) + (1 | Participant)* |
| Model 7 | *Distance ~ 1 + Protective Equipment × COVID-19 Test Result + Participant's Gender × Other Avatar's Gender + Perceived Severity of the situation in the Country + Virtual Contact + Physical Contact + Perceived Vulnerability to Disease + Individualizing Moral Foundation + Binding Moral Foundation + (1 | Country) + (1 |Participant)* |
| Model 8 | *Distance ~ 1 + Protective Equipment × COVID-19 Test Result + Participant's Gender × Other Avatar's Gender + Perceived Severity of the situation in the Country + Virtual Contact + Physical Contact + Perceived Vulnerability to Disease + Individualizing Moral Foundation + Binding Moral Foundation + Quarantine's Justification + (1 | Country) + (1 |Participant)* |
| Model 9 | *Distance ~ 1 + Protective Equipment × COVID-19 Test Result + Participant's Gender × Other Avatar's Gender + Perceived Severity of the situation in the Country + Virtual Contact + Physical Contact + Perceived Vulnerability to Disease + Individualizing Moral Foundation + Binding Moral Foundation + Quarantine's Justification + Altruism + (1 | Country) + (1 |Participant)* |
| Model 10 | *Distance ~ 1 + Protective Equipment × COVID-19 Test Result + Participant's Gender × Other Avatar's Gender x Participant's Sexual Orientation + Perceived Severity of the situation in the Country + Virtual Contact + Physical Contact + Perceived Vulnerability to Disease + Individualizing Moral Foundation + Binding Moral Foundation + Quarantine's Justification + Altruism + (1 | Country) + (1 |Participant)* |

distributions and identify potential errors/anomalies, and (ii) identify which models to adopt for a better fit of our data.

To account for the nested structure of our sample, that is, participants nested within countries, multilevel modeling was used [50]. Multilevel models of increasing complexity were fitted in order to select the most accurate one. A list of all models can be found in Table 2. The starting point was a "Primary" Model, which included the main effects of COVID-19 Test Result, Protective Equipment, Participant's Gender and Other Avatar's Gender. The interactions between COVID-19 Test Result and Protective Equipment and between Participant's and Other Avatar's Gender were also included as predictors. All models presented here included the intercept over Participants and Countries as random effects (see Table 1).

All continuous predictors were mean-centered. The two ordinal scales Virtual and Physical Contact were dichotomized into "Frequent" (including "Often" and "Always") and "Infrequent" (including "Never," "Rarely," and "Sometimes"), while Sexual Orientation was categorized into "Heterosexual" (including Kinsey Scale's 1–3 scores), "Non Heterosexual" (including Kinsey Scale's 4–7 scores) and "No socio-sexual contacts or reactions" (Kinsey Scale's 8 score). Non-informative, normally-distributed priors ($M = 0$, $SD = 1000$) were applied to all models, on all population-level effects and $t$- distributed priors ($df = 3$, $M = 0$, $SD = 28$) on the intercept and on the group-level effects. Use of non-informative prior prevents results from being biased toward alternative hypotheses [51] and respects the Laplacian principle of indifference [52]. All models were fitted using four independent Markov chains. Each chain had 30,000 iterations, the first 15,000 of which were warm-up. This led to a total of 60,000 post warm-up posterior samples for inference. According to standard convergence diagnostics [53], all models converged (Rhat < 1.05) with sufficient precision (effective sample size > 1000).

The models were compared through approximate leave-one-out cross-validation and using Pareto-smoothed importance sampling (PSIS-LOO [30]), which estimates out-of-sample predictive accuracy adopting within-sample fits. Model 5 had the best predictive accuracy and included the same structure of the Primary model plus the main effects of Perceived Severity

**Table 3. Model comparison via leave-one-out cross-validation.**

| Model | ELPD-diff | SE-diff | weight |
|---|---|---|---|
| Model 5 | 0 | 0 | 0.14 |
| Model 3 | -0.1 | 0.3 | 0.13 |
| Model 1 | -0.2 | 0.5 | 0.11 |
| Model 6 | -0.3 | 0.1 | 0.11 |
| Model 2 | -0.3 | 0.5 | 0.10 |
| Model 4 | -0.3 | 0.1 | 0.10 |
| Model 8 | -0.3 | 0.4 | 0.10 |
| Model 7 | -0.4 | 0.3 | 0.09 |
| Model 9 | -0.5 | 0.4 | 0.08 |
| Model 10 | -3.9 | 0.7 | 0 |
| Model 0 | -2200.3 | 58.1 | 0 |

We report the differences in the point estimates (ELPD-diff) and standard errors of the difference (SE-diff) of the expected log pointwise predictive density (ELPD). The values in the ELPD-diff and SE-diff columns of the returned matrix are computed by making pairwise comparisons between each model and the model with the largest ELPD (the model in the first row). ELPD indicates the predictive performance of the model. Model weights are calculated via stacking of the predictive distributions: The method combines all models by maximizing the leave-one-out predictive density of the combination distribution.

of the situation in the country, Virtual and Physical Contact, and Perceived Vulnerability to Disease (see Table 3).

## Final model

To visualize the probability of direction of the effect for each parameter included in the study, see Fig 2. The summary of the model is reported in Table 4. Table 5 presents contrasts between all levels of Protective Equipment and COVID-19 Test Result; a graphical representation of this interaction can be found in Fig 3. Contrasts between all levels of Participant's Gender and Other Avatar's Gender are reported in Table 6.

Analysis of the final model (Model 5) focused on posterior contrasts between all levels of categorical predictors and the slope of continuous predictors. In order to quantify the uncertainty and magnitude of effects, we computed the 95% highest density interval (HDI). Any parameter value inside the HDI has higher probability density than any parameter value outside the HDI [54]. However, the credible interval is conditional on $H1$ being true and quantifies the strength of an effect, assuming it is present [55]. To quantify evidence for presence or absence of the effects, we computed the Bayes factors [42]. The BF quantifies the relative predictive performance of two rival hypotheses, and represents the degree to which data require a change in beliefs concerning the relative plausibility hypotheses [55]. A common rule of thumb is the following: $BF_{10} > 3$ indicates support for the alternative hypothesis and $BF_{10} < 0.333$ suggests support for the null hypothesis [55].

We found strong evidence that the preferred IPD was shorter for the Negative-diagnosed avatar in comparison with the Positive-diagnosed (estimate = -27.90, HDI [-28.65, -27.11], $BF_{10}$ = 3.526e+97, see Fig 3) and with the Unknown-diagnosed avatar (estimate -11.37, HDI [-12.13, -10.59], $BF_{10}$ = 9.795e+42). The preferred IPD was larger for the Positive-diagnosed compared to the Unknown-diagnosed (estimate 16.52, HDI [15.76, 17.30], $BF_{10}$ = 1.458e+47). The preferred IPD was shorter for the Worn Protective Equipment condition compared to the Not Worn Protective Equipment condition (estimate = -6.58, HDI [-7.67, -5.50], $BF_{10}$ = 1.08E+11). This effect was present when considering the Negative-diagnosed avatar, (estimate = -6.58, HDI [-7.67, -5.50], $BF_{10}$ = 2.68E+09), the Unknown-diagnosed avatar (estimate = -6.41, HDI [-7.48, -5.30], $BF_{10}$ = 8.81E+08) and Positive-diagnosed avatar (estimate = -7.60, HDI [-8.69, -6.52], $BF_{10}$ = 1.08E+13).

We also found non-zero effects for Participant's Gender and Virtual Contact. However, in both cases the Bayes factor indicated moderate evidence in support of the null hypothesis: the preferred IPD was larger for women compared to men (estimate = 5.08, HDI [0.08, 10.06], $BF_{10}$ = 0.02) and for participants who had Infrequent Virtual Contact during the two weeks prior to participation compared to participants who had Frequent Virtual Contact (estimate = -8.71, HDI [-14.23, -3.07], $BF_{10}$ = 0.28). No other credible effects were found.

## Discussion

Interpersonal distance of at least 1 m is a fundamental measure of containment for the spreading of SARS-CoV-2. Adherence to this measure represents a dramatic change from people's behavior under normal circumstances. As of now, the dispositional and situational factors that impact adherence to this rule are under-investigated. In this study, we explored the role of protective equipment, actual risk of infection, perceived vulnerability, severity of the situation, physical and virtual contacts, morality, attitudes toward quarantine, and prosocial tendencies in the regulation of IPD during the COVID-19 outbreak. Using a model selection approach, we aimed to identify the most relevant variables to predict IPD behavior. In line with previous studies that investigated the distance maintained from infected individuals [17, 18], we found

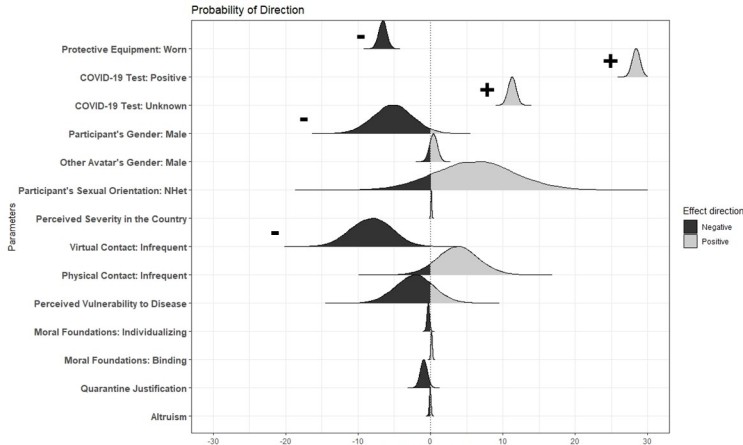

**Fig 2. Probability of direction and the magnitude of the effect for each predictor included in the study.** The y-axis indicates the predictors and the x-axis indicates the possible parameter values. The color indicates the direction of the effect: black stands for a negative direction (reduction of IPD), while gray represents a positive direction (enlargement of IPD). The effect of the parameters included in the final model whose HDI are completely outside of zero are marked with "-" (if the direction of the effect is negative) or a "+" (if the direction of the effect is positive). The interaction between COVID-19 Test Result and Protective Equipment and the interaction between Participant's Gender and Other Avatar's Gender are better explained by the contrasts between all levels of the factors (COVID-19 Test Result: Protective Equipment see Table 4 and Fig 3; Participant's Gender: Other Avatar's Gender see Table 5).

strong evidence that providing information regarding a positive COVID-19 test result increased IPD. In particular, we found a continuous increase in the space that participants put

**Table 4. Summary of the final model.**

| Parameter | Median | 95% HDI | BF$_{10}$ | ESS | R̂ |
|---|---|---|---|---|---|
| Intercept | 33.61 | [28.11, 39.25] | 8.33E+10 | 9319 | 1 |
| *Protective Equipment* | -6.58 | [-7.67, -5.50] | 1.08E+11 | 33152 | 1 |
| Worn v. Not Worn | | | | | |
| *COVID-19 test result* | -27.90 | [-28.66, -27.11] | 3.526e+97 | 35176 | 1 |
| Negative v. Positive | | | | | |
| Negative v. Unknown | -11.37 | [-12.14, -10.59] | 9.795e+42 | 35998 | 1 |
| Positive v. Unknown | 16.52 | [15.76, 17.30] | 1.458e+47 | 35765 | 1 |
| *Participant's Gender* | 5.08 | [0.08, 10.06] | 0.02 | 6970 | 1 |
| Female v. Male | | | | | |
| *Other Avatar's Gender* | 0.40 | [-0.39, 1.22] | 6.759e-04 | 39129 | 1 |
| Male v. Female | | | | | |
| *Perceived Severity of the situation in the Country* | 0.10 | [-0.01, 0.22] | 2.773e-04 | 9688 | 1 |
| *Virtual contact* | -8.71 | [-14.23, -3.07] | 0.28 | 10351 | 1 |
| Infrequent v. Frequent | | | | | |
| *Physical contact* | 5.04 | [-0.38, 10.28] | 0.015 | 9056 | 1 |
| Infrequent v. Frequent | | | | | |
| *Perceived Vulnerability to Disease* | -1.04 | [-5.99, 3.85] | 0.003 | 8344 | 1 |

For all categorical predictors, we report the contrasts between each level of the factor; for the continuous predictors, we report the regression coefficient which represents the change in the outcome resulting from a unit change in the predictor. Results are described by means of the Median, the 95% HDI (Highest Density Interval) and the BF (Bayes Factor). A BF greater than 3 indicates support for the alternative hypothesis, while the HDI quantifies the magnitude of the effect and its uncertainty.

**Table 5. Contrasts between all levels of protective equipment and COVID-19 test result.**

| Protective Equipment: COVID-19 test result Contrasts | Median | 95% HDI | BF$_{10}$ |
|---|---|---|---|
| Negative: Worn v. Not Worn | -6.58 | [-7.67, -5.50] | 2.68E+09 |
| Unknown: Worn v. Not Worn | -6.41 | [-7.48, -5.30] | 8.81E+08 |
| Positive: Worn v. Not Worn | -7.60 | [-8.69, -6.52] | 1.08E+13 |
| Worn: Negative v. Unknown | -11.46 | [-12.55, -10.38] | 6.6E+19 |
| Worn: Positive v. Unknown | 15.94 | [14.84, 17.01] | 1.72E+79 |
| Worn: Negative v. Positive | -27.39 | [-28.48, -26.30] | 1.05E+70 |
| Not Worn: Negative v. Unknown | -4.88 | [-5.96, -3.78] | 645871.4 |
| Not Worn: Positive v. Unknown | 17.13 | [16.03, 18.21] | 1.19E+31 |
| Not Worn: Negative v. Positive | -28.42 | [-29.49, -27.31] | 2.14E+78 |
| Worn x Negative v. Not Worn x Positive | -35.00 | [-36.09, -33.90] | 1.34E+86 |
| Not Worn x Negative v. Worn x Positive | -20.81 | [-21.91, -19.75] | 7.18E+53 |
| Worn x Negative v. Not Worn x Unknown | -17.87 | [-18.94, -16.76] | 1.27E+44 |
| Not Worn x Negative v. Not Worn x Unknown | -11.28 | [-12.39, -10.22] | 7.35E+24 |
| Worn x Positive v. Not Worn x Unknown | 9.52 | [8.44, 10.60] | 5.87E+20 |
| Not Worn x Positive v. Worn x Unknown | 23.53 | [22.43, 24.61] | 1.4E+63 |

between themselves and another person, with the shortest distance reported in association with a negative-tested individual, a medium distance observed when the other individual had an unknown-test result, and a maximum distance when the other person tested positive to COVID-19. These results may reflect the notion of "behavioral immune system" [56], according to which humans use behavioral avoidance of disease-causing objects and people as a disease-management strategy. The evidence that our three different conditions are associated with a continuously increasing space between participants and the another person suggests that the purported behavioral immune system may be regulated by a probabilistic inference about risk [57]: the higher the perceived risk, the larger the IPD. Indeed, when participants were not informed about the other person's COVID-19 test result (i.e., Unknown condition) they might have relied on the conviction that the other had a 50% chance of being infected, thus placing themselves between the more extreme conditions (where a 0% risk is associated with the Negative condition and an estimated 100% risk to the Positive one).

We also found strong evidence that interacting with someone who was wearing protective equipment was associated with reduced IPD. This result is consistent with the findings of other studies conducted during the COVID-19 pandemic that used different methodologies [19, 20]. Specifically, Iachini and colleagues [20] used an 8-point Likert scale (ranging from 1 = *0.5 m* to 8 = *4 m*) and found that the comfort-distance from others wearing a mask was shorter than the one from others without a mask. Cartaud and colleagues [19] used characters showing either positive, neutral, or negative facial expression or wearing a mask (which was always associated to a neutral facial expression). Characters were presented at different fixed distances from participants. Results showed that shorter IPD was judged as more appropriate for the characters wearing a mask compared to the other conditions; these characters were also perceived as more trustworthy. It has been suggested that the sight of a mask could induce a

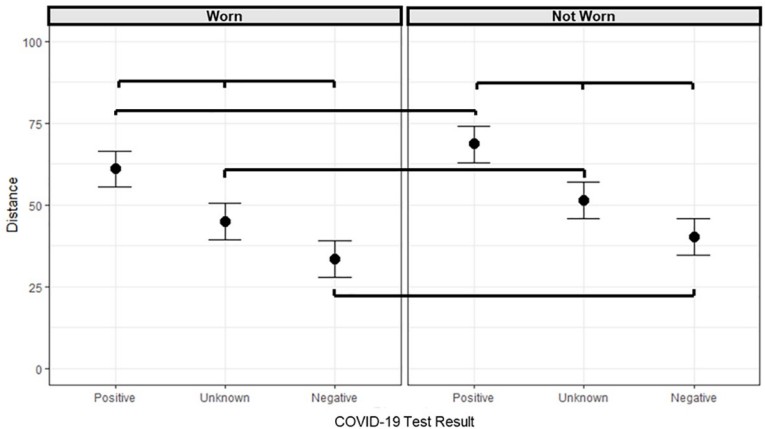

**Fig 3. Parameter estimates from the interaction between COVID-19 test result and protective equipment.** The central dot indicates the posterior median and the whiskers indicate the lower and upper limits of the 95% HDI. The contrasts for which the 95% HDI does not include zero and for which $BF_{10} > 3$ (support for the alternative hypothesis) are connected with lines.

feeling of safety that facilitates a reduction of IPD. Enactment of this tendency in real-life situations may constitute a potential threat, since the use of protective equipment alone is not enough to prevent SARS-CoV-2 from spreading [21]. Therefore, one question that remained unanswered is whether the reduction of IPD in response to the sight of protective equipment also occurs when there is an actual risk of infection. Importantly, at variance with the previous studies, we observed that the effect of mask wearing is present not only when participants are not provided with information regarding the other person's COVID-19 test result, but also when they are provided with information of a positive diagnosis. In public settings, allowing a shorter IPD to a person wearing protective equipment may create conditions for further transmission of the new coronavirus.

Among other predictors, Virtual Contact had a non-zero effect, meaning that frequent virtual contact led to larger IPD compared to infrequent virtual contact. However, it is worth noting that the Bayes factor analysis did not reveal support for this effect. According to the contact hunger hypothesis, the type of social isolation individuals may experience during the pandemic could lead to an enhanced need for physical contact [22]. Indeed, affiliation and contact-seeking are core responses to perceived danger [58, 59], and this may happen even in cases where contact itself is a threat, as in infectious diseases. It is possible that engaging in frequent virtual contacts may have modulated this evolutionary drive, leading to lower motivation for interpersonal connection in comparison to those who experienced infrequent virtual contact. Future studies should systematically investigate the effect of virtual contact's quantity and quality in modulating IPD during social isolation.

**Table 6. Contrasts between all levels of participant's and other avatar's gender.**

| Participant's Gender: Other Avatar's Gender Contrasts | Median | 95% HDI | $BF_{10}$ |
|---|---|---|---|
| Female Participants: Female Avatar v. Male Avatar | 0.36 | [-1.35, 0.61] | 4.557e-04 |
| Male Participants: Female Avatar v. Male Avatar | -0.41 | [-1.23, 0.39] | 6.761e-04 |
| Female Avatar: Male Participants v. Female Participants | -5.09 | [-10.06, -0.08] | 0.019 |
| Male Avatar: Male Participants v. Female Participants | -5.04 | [-10.04, -0.11] | 0.013 |
| Male Participants x Female Avatar v. Female Participants x Male Avatar | -5.45 | [-10.40, -0.43] | 0.015 |
| Female Participants, Female Avatar v. Male Participants, Male Avatar | 4.67 | [-0.19, 9.72] | 0.01 |

Overall, women kept a larger distance from others compared to men, although Bayes factor analysis did not show decisive support for this effect. Women, indeed, tend to exhibit more defensive behavior during interactions with strangers [5, 60]. Interestingly, this result, which requires further investigation, is in line with research on gender differences in the pandemic response, which showed that men's belief of being gravely affected by COVID-19 is reduced with respect to women [61]. Additionally, men appear to be less likely to comply with preventive behaviors [62]. Contrary to previous evidence [13, 15], participant's sexual orientation did not modulate the gender differences, suggesting that, in the context of a pandemic, its relevance may be reduced.

It should be noted that, because this study is based on hypothetical choices, we cannot provide a conclusive answer to the question of how people regulate IPD during the spread of an infectious disease. Although in the domain of physical distancing there is evidence that self-report measures are correlated with actual behavior [61], it is difficult to rule out the influence of social desirability bias and thus the present findings must be replicated in a more ecological context.

Moreover, future studies are needed to clarify whether the reduction of IPD following the sight of worn protective equipment is present across different public contexts (i.e., hospitals, supermarkets) and whether this effect is dependent on the social encoding of the other person. Finally, the results of our study must be considered in light of recent neuroscientific evidence [63], which shows that IPD regulation may be rooted in the peripersonal space representation (the multisensory motor area within which it is possible to reach and interact with objects [64]). Indeed, Vieira and colleagues [65] showed that distance from other organisms (conspecifics or not) is regulated by a network that includes the midbrain periaqueductal gray (a region sensitive to threat proximity and involved in defensive behaviors) and frontoparietal structures representing peripersonal space. Future neuroimaging studies may allow to investigate whether the reduction of IPD associated to seeing another person wearing a mask reflects a modulation of the activity in the above network, therefore supporting the hypothesis of a reduced perceived threat.

## Supporting information

**S1 File. Supplemental information of: A Bayesian approach to reveal the key role of mask wearing in modulating projected interpersonal distance during the first COVID-19 outbreak.**
(DOCX)

## Author Contributions

**Conceptualization:** Matteo P. Lisi, Marina Scattolin, Martina Fusaro, Salvatore Maria Aglioti.

**Data curation:** Matteo P. Lisi, Marina Scattolin.

**Formal analysis:** Matteo P. Lisi.

**Funding acquisition:** Matteo P. Lisi, Martina Fusaro, Salvatore Maria Aglioti.

**Investigation:** Matteo P. Lisi, Marina Scattolin, Martina Fusaro.

**Methodology:** Matteo P. Lisi, Marina Scattolin, Martina Fusaro.

**Software:** Matteo P. Lisi, Marina Scattolin.

**Supervision:** Salvatore Maria Aglioti.

**Visualization:** Matteo P. Lisi.

**Writing – original draft:** Matteo P. Lisi.

**Writing – review & editing:** Matteo P. Lisi, Marina Scattolin, Martina Fusaro, Salvatore Maria Aglioti.

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
