## [Decision Letter · Decision Letter 0]

26 Mar 2021

PONE-D-21-03526

A Bayesian approach to reveal the key role of mask wearing in modulating interpersonal distance during the first COVID-19 outbreak

PLOS ONE

Dear Dr. Lisi,

Thank you for submitting your manuscript to PLOS ONE. After careful consideration, we feel that it has merit but does not fully meet PLOS ONE’s publication criteria as it currently stands. Therefore, we invite you to submit a revised version of the manuscript that addresses the points raised during the review process.

Please find below the reviewers' comments, as well as those of mine.

We look forward to receiving your revised manuscript.

Kind regards,

Valerio Capraro

Academic Editor

PLOS ONE

Journal Requirements:

2. If materials, methods, and protocols are well established, authors may cite articles where those protocols are described in detail, but your submission should include sufficient information to be understood independent of these references (https://journals.plos.org/plosone/s/submission-guidelines#loc-materials-and-methods).

Additional Editor Comments:

I have now collected three reviews from three experts in the field. All reviewers think that the topic of the paper is important, but their final judgment is split: two recommend major revision, one recommends rejection. The negative review is based on the observation that the method is inappropriate for answering the question, because it is based on hypothetical choices, while the first field experiments on the topic are already coming out. While I understand this reviewer's objection, at the same time I think that also this approach is valuable, especially because there is also some evidence that self-report measures are correlated to actual behavior, at least in the domain of physical distancing (Gollwitzer et al. 2020). Moreover, the field experiment mentioned by the reviewer is a working paper. Therefore, I have decided to follow the majority and invite you to revise your work. Needless to say that all comments must be addressed. In particular, I expect you to try to improve your writing in order to accomodate also the negative reviewer: make explicit the fact that your results are based on hypothetical choices and discuss the limitations of this approach in details and stress the need for future work. Perhaps also the title should be changed in order to make clear that your measure is hypothetical Also, I would like to add a couple more comments regarding the literature review, which I found to be incomplete: (i) the "perspective article" on what social and behavioural science can do to support pandemic response, published by Van Bavel et al in Nature Human Behaviour, could be a useful general reference; (ii) I was surprised to see that you did not review the emerging literature on the role of prosociality on pandemic response. I am aware of at least seven published papers looking at this (Banker et al. 2020; Bilancini et al. 2020; Capraro & Barcelo, 2020; Campos-Mercade et al. 2020; Heffner et al. 2020; Lunn et al. 2020; Pfattheicher et al. 2020).

I am looking forward for the revision.

References

Banker, S., & Park, J. (2020). Evaluating prosocial COVID-19 messaging frames: Evidence from a field study on Facebook. Judgment and Decision Making, 15(6), 1037-1043.

Bilancini E, Boncinelli L, Capraro V, Celadin T, Di Paolo R (2020) The effect of norm-based messages on reading and understanding COVID-19 pandemic response governmental rules. Journal of Behavioral Economics for Policy 4, Special Issue 1, 45-55.

Capraro, V., & Barcelo, H. (2020). The effect of messaging and gender on intentions to wear a face covering to slow down COVID-19 transmission. Journal of Behavioral Economics for Policy, 4, Special Issue 2, 45-55.

Campos-Mercade, P., Meier, A. N., Schneider, F. H., & Wengström, E. (2021). Prosociality predicts health behaviors during the COVID-19 pandemic. Journal of Public Economics, 195, 104367.

Heffner, J., Vives, M. L., & FeldmanHall, O. (2020). Emotional responses to prosocial messages increase willingness to self-isolate during the COVID-19 pandemic. Personality and Individual Differences, 170, 110420.

Lunn, P. D., Timmons, S., Barjaková, M., Belton, C. A., Julienne, H., & Lavin, C. (2020). Motivating social distancing during the Covid-19 pandemic: An online experiment. Social Science & Medicine, 113478.

Pfattheicher, S., Nockur, L., Böhm, R., Sassenrath, C., & Petersen, M. B. (In press). The emotional path to action: Empathy promotes physical distancing during the COVID-19 pandemic. Psychological Science.

Van Bavel, J. J., et al. (2020). Using social and behavioural science to support COVID-19 pandemic response. Nature Human Behaviour, 4, 460-471.

Reviewers' comments:

Reviewer's Responses to Questions

**Comments to the Author**

1. Is the manuscript technically sound, and do the data support the conclusions?

Reviewer #1: Yes

Reviewer #2: Yes

Reviewer #3: Yes

2. Has the statistical analysis been performed appropriately and rigorously? 

Reviewer #1: Yes

Reviewer #2: I Don't Know

Reviewer #3: Yes

3. Have the authors made all data underlying the findings in their manuscript fully available?

Reviewer #1: Yes

Reviewer #2: No

Reviewer #3: Yes

4. Is the manuscript presented in an intelligible fashion and written in standard English?

Reviewer #1: Yes

Reviewer #2: Yes

Reviewer #3: Yes

5. Review Comments to the Author

Reviewer #1: I think this is an important question, but ultimately the methods are inappropriate to answer it.

There is better evidence from real-world field experiments along similar lines:

https://papers.ssrn.com/sol3/papers.cfm?abstract_id=3641367

Given this, extrapolating from hypothetical choices on laboratory screens seems unnecessary given the many possible differences between this and the real-world. Foremost among them is that respondents are answering what they think experimenters want rather than what they would actually do in practice.

Reviewer #2: The manuscript focuses on a relevant topic concerning a very actual issue as it investigates the effect of protective equipment and knowledge in COVID-19 test results on the regulation of interpersonal distances (IPD). This work is very interesting and important in the context of the COVID-19 pandemic as well as in pandemic context in general. However, I have several concerns about different aspects of the study that I report bellow:

1. I cannot access the data and the codes with the link provided by the authors (which seems to be down). Therefore, I cannot take a look neither at the statistical code nor at the data. This is the reason why I answered “I don’t know” and “No” to question 2 and 3 respectively. This is also the reason why I am recommending major revisions. However, the arguments related to the choice of the statistical analysis in the text are clear and the statistical analysis used is appropriate.

2. In the introduction section, even if Hayduk is mentioned in the text, I think Hall should also be cited, at least when introducing proxemics.

3. The experimental method may not be the best to use (see Hayduk, 1983), although it is quite understandable to use given the sanitary context. As the task is particularly sensitive to expectation bias, how the authors could rule out this potential confound? I noticed the authors mentioned this limitation in the discussion section.

4. P3, l.43. “Although previous research has helped define IPD under normal circumstances, insights on which factors influence the regulation of IPD during a pandemic are lacking”. This sentence is contradictory with the following one (citing studies focusing on different factors influencing IPD during a pandemic).

5. In Table 2:

5.1. Random effects are specified twice in every model except Model 0, why?

5.2. Can the authors explain why they added 2 or 3 variables between Model 2 and 3, 3 and 4 and 4 and 5, rather than adding one variable at the time for the LOO comparison?

6. The authors could write directly in the text that Model 4 is the final model in order to improve the clarity of the manuscript.

7. Table 4: There is a typo: negative sign outside of the hook in the Negative v. Unknown comparison.

8. Table 4 and 5: I think there is a mistake when reporting the results. the Median and 95% HDI of the Negative: Worn v. Not Worn comparison (Table 5) are equal to the Median and 95% HDI of the Protective Equipment comparison Worn v. Not Worn (Table 4, -6.58 [-7.67, -5.50]).

9. I was particularly interested by the results regarding the COVID-19 test results variable (first time reported to my knowledge) and especially by the results of the Unknown condition as it suggests that individuals think about risk in a “probabilistic” way (50% chance the individual is sick: medium distance). I was a little disappointed the authors did not develop more in the discussion section about those results. I think it is a key point of this research.

Reviewer #3: In a web-based experimental study conducted during the first pandemic wave (mid-April 2020), the authors tested preferred interpersonal distance with confederates. The variables explored concerned the role of protective equipment, actual risk of infection, perceived vulnerability, severity of the situation, physical and virtual contacts, morality, attitudes toward quarantine, and prosocial tendencies in the regulation of IPD during the COVID-19 outbreak. The test was based on the Interpersonal Visual Analogue Scale (IVAS), adapted to the present study. Based on Bayesian analysis approach the authors found evidence in favor of a reduction of interpersonal distance towards individuals wearing protective equipment and who tested negative to COVID- 19. Shorter interpersonal distances were also found with confederates wearing protective gear, even when COVID-19 test result was unknown or positive. Individual differences did not modulate significantly interpersonal distances. The protective equipment-related regulation of interpersonal distance may reflect an underestimation of perceived vulnerability to infection. Consequences in terms of collective health-safety measures communication are discussed.

The aim of this study was to investigate whether being at risk of infection or having specific personal characteristics modulates IPD. The effects of variables were tested using a model selection approach embedded in a Bayesian analysis approach. The study was a web-based experimental study, but provided interesting results. I have only few comments relating to the state of the art in the research domain, and the analysis of the data, exposed hereafter.

Introduction

When stating “handwashing and use of face masks have been widely adopted in conjunction with maintaining interpersonal distances of at least 1.5 m”, this is in fact dependent on the country (see for instance https://theprint.in/theprint-essential/1m-1-5m-2m-the-different-levels-of-social-distancing-countries-are-following-amid-covid/449425/)

When indicating “Research in proxemics, the study of interpersonal spatial behavior …” you should quote Hall (1966), who is at the origin of the research field.

When mentioning “IPD is shaped by situational factors such as social threat…” you should mention the demonstration made by Cartaud et al. (2018, Frontiers Psychology),

When indicating “IPD appears to be automatically regulated according to distance-related feelings of personal comfort », you should quote and perhaps discuss the recent model proposed by Coello & Cartaud (2021, Frontiers in Human Neuroscience).

Data analysis

It is not clear why sometimes the model includes one additional variable (Model 2 for instance) and sometimes two (Model 3 for instance).

Please, when discussing the Final Model, indicates that this refers to Model 4.

When discussing Fig 2, please indicate what negative/positive directions correspond to in terms of the measure.

I suggest to indicate which difference is significant in Figure 2. For instance, Gender effect (participants) is significant but this is not clear from the representation used in Figure 2.

You indicate that « IPD was shorter for women compared to men », but Figure 2 shows the opposite.

Discussion

It can be indicated that people wearing a mask are also perceive as more trustworthy (Cartaud et al., 2020).

When indicating “Among the other predictors, only Virtual Contact had a non-zero effect”, this is not in agreement from the information provided in the data analysis section where it is mentioned a “non-zero effects for Participant’s Gender and Virtual Contact”.

The Gender effect is not discussed and seems opposite to what is usually reported in the literature (for instance, Iachini et al., 2016, Journal of Environmental Psychology)

When concluding that “the results of our study must be considered in light of recent neuroscientific evidence, which shows that the distance from other organisms (conspecific or not) is regulated by a common network … representing peripersonal space”. I am not sure about what means “common network” in this sentence. Furthermore, on this issue I recommend the authors to refer to the recent paper by Coello & Cartaud (2021, Frontiers in Human Neuroscience), who propose a new model to account for the relation between IPD and peripersonal space.

6. PLOS authors have the option to publish the peer review history of their article (what does this mean?). If published, this will include your full peer review and any attached files.

Reviewer #1: No

Reviewer #2: **Yes: **Alice Cartaud

Reviewer #3: No

---

## [Author Response · Author response to Decision Letter 0]

6 May 2021

We enclose a point-by-point reply to the comments of each reviewer (our responses in bold; additions to the revised MS in bold&red).

Reviewer #1

Ref1#1. I think this is an important question, but ultimately the methods are inappropriate to answer it.

There is better evidence from real-world field experiments along similar lines:

https://papers.ssrn.com/sol3/papers.cfm?abstract_id=3641367

Given this, extrapolating from hypothetical choices on laboratory screens seems unnecessary given the many possible differences between this and the real-world. Foremost among them is that respondents are answering what they think experimenters want rather than what they would actually do in practice.

ARRef1#1: 

We acknowledge this approach is generally limited by desirability bias and that our results are based on hypothetical choices. However, our decision to employ a projective method was motivated by: i) sanitary reasons, and ii) the possibility of having full control over the manipulated variables. Also, we believe that had their responses been biased by social desirability, we would expect participants to predict keeping a similar distance from avatars associated with Positive Covid-19 Test Result and from those labelled as Unknown. In fact, the latter condition represented the most realistic situation, as people are not usually aware of the other’s health status and therefore are supposed to maintain a safe distance a priori. Instead, we found that the predicted interpersonal distance from avatars associated with Unknown test results was shorter than the one from avatars showing a Positive diagnosis. This suggests that participants may indeed have provided responses based on their subjective evaluation, and thus they did not just apply the recommended rules of conduct. Additionally, it is worth noting that projective measures of physical distancing have been found to correlate with actual behavior (Gollwitzer et al., 2020). 

Nevertheless, to discuss the limitations of our approach, we added the following paragraph to the Discussion section, which now reads (page 26-27, lines 410-415): “It should be noted that, because this study is based on hypothetical choices, we cannot provide a conclusive answer to the question of how people regulate IPD during the spread of an infectious disease. Although in the domain of physical distancing there is evidence that self-report measures are correlated with actual behavior [61], it is difficult to rule out the influence of social desirability bias and thus the present findings must be replicated in a more ecological context.” 

We also decided to modify the title by adding the word “projected”, which we believe better describes the hypothetical nature of our task. Thus, the title now reads as follows: “A Bayesian approach to reveal the key role of mask wearing in modulating projected interpersonal distance during the first COVID-19 outbreak”.

Reviewer #2: The manuscript focuses on a relevant topic concerning a very actual issue as it investigates the effect of protective equipment and knowledge in COVID-19 test results on the regulation of interpersonal distances (IPD). This work is very interesting and important in the context of the COVID-19 pandemic as well as in pandemic context in general. However, I have several concerns about different aspects of the study that I report bellow:

We thank the Reviewer for his/her positive evaluation of our work. We addressed his/her points below. 

Ref2#1. I cannot access the data and the codes with the link provided by the authors (which seems to be down). Therefore, I cannot take a look neither at the statistical code nor at the data. This is the reason why I answered “I don’t know” and “No” to question 2 and 3 respectively. This is also the reason why I am recommending major revisions. However, the arguments related to the choice of the statistical analysis in the text are clear and the statistical analysis used is appropriate

ARRef2#1: We apologize for this mistake. The dataset is now available at the following link: https://data.mendeley.com/datasets/jw3sbz2nkv/1 .

Ref2#2. In the introduction section, even if Hayduk is mentioned in the text, I think Hall should also be cited, at least when introducing proxemics.

ARRef2#2: We thank the Reviewer for suggesting this. We now cited Hall when defining interpersonal distance (page 3, lines 41-43): “Research in proxemics, the study of interpersonal spatial behavior [7,8], has defined interpersonal distance (IPD) as the separation zone that individuals keep between themselves and others [9].”

Ref2#3. The experimental method may not be the best to use (see Hayduk, 1983), although it is quite understandable to use given the sanitary context. As the task is particularly sensitive to expectation bias, how the authors could rule out this potential confound? I noticed the authors mentioned this limitation in the discussion section.

ARRef2#3: We thank the Reviewer for this comment, which is similar to an issue raised by Reviewer 1. 

We acknowledge this approach is generally limited by desirability bias and that our results are based on hypothetical choices. However, our decision to employ a projective method was motivated by: i) sanitary reasons, and ii) the possibility of having full control over the manipulated variables. Also, we believe that had their responses been biased by social desirability, we would expect participants to predict keeping a similar distance from avatars associated with Positive Covid-19 Test Result and from those labelled as Unknown. In fact, the latter condition represented the most realistic situation, as people are not usually aware of the other’s health status and therefore are supposed to maintain a safe distance a priori. Instead, we found that the predicted interpersonal distance from avatars associated with Unknown test results was shorter than the one from avatars showing a Positive diagnosis. This suggests that participants may indeed have provided responses based on their subjective evaluation, and thus they did not just apply the recommended rules of conduct. Additionally, it is worth noting that projective measures of physical distancing have been found to correlate with actual behavior (Gollwitzer et al., 2020). 

Nevertheless, to discuss the limitations of our approach, we added the following paragraph to the Discussion section, which now reads (page 26-27, lines 410-415): “It should be noted that, because this study is based on hypothetical choices, we cannot provide a conclusive answer to the question of how people regulate IPD during the spread of an infectious disease. Although in the domain of physical distancing there is evidence that self-report measures are correlated with actual behavior [61], it is difficult to rule out the influence of social desirability bias and thus the present findings must be replicated in a more ecological context.” 

We also decided to modify the title by adding the word “projected”, which we believe better describes the hypothetical nature of our task. Thus, the title now reads as follows: “A Bayesian approach to reveal the key role of mask wearing in modulating projected interpersonal distance during the first COVID-19 outbreak.”

Ref2#4. P3, l.43. “Although previous research has helped define IPD under normal circumstances, insights on which factors influence the regulation of IPD during a pandemic are lacking”. This sentence is contradictory with the following one (citing studies focusing on different factors influencing IPD during a pandemic). 

ARRef2#4: We thank the Reviewer for this comment, and we apologize for not being clear. We now corrected the sentence (page 3, lines 46-48): “Although previous research has largely investigated IPD under regular circumstances, much less is known about the factors influencing the regulation of IPD during the spread of infectious diseases.”

Ref2#5. 5. In Table 2:

5.1. Random effects are specified twice in every model except Model 0, why?

5.2. Can the authors explain why they added 2 or 3 variables between Model 2 and 3, 3 and 4 and 4 and 5, rather than adding one variable at the time for the LOO comparison?

ARRef2#5: 

5.1. We thank the Reviewer for identifying this typo. We now reported the correct formula for each model in Table 2. 

5.2. We have now reported a new analysis in which just one variable differentiates one model from the following one. 

 Table 2. Formulas for each model

Ref2#6. The authors could write directly in the text that Model 4 is the final model in order to improve the clarity of the manuscript.

ARRef2#6: We thank the Reviewer for suggesting this. In the new analysis, Model 5 is the final one. We added the following sentence to page 14-15, lines 264-267: “Model 5 had the best predictive accuracy and included the same structure of the Primary model plus the main effects of Perceived Severity of the situation in the country, Virtual and Physical Contact, and Perceived Vulnerability to Disease (see Table 3).” For the sake of clarity, we also specified this at page 18, line 284: “Analysis of the final model (Model 5) focused on posterior contrasts between all levels of categorical predictors and the slope of continuous predictors.”

Ref2#7. Table 4: There is a typo: negative sign outside of the hook in the Negative v. Unknown comparison. 

ARRef2#7: We thank the Reviewer for highlighting this typo which we have now corrected.

Ref2#8. Table 4 and 5: I think there is a mistake when reporting the results. the Median and 95% HDI of the Negative: Worn v. Not Worn comparison (Table 5) are equal to the Median and 95% HDI of the Protective Equipment comparison Worn v. Not Worn (Table 4, -6.58 [-7.67, -5.50]).

ARRef2#8: We thank the Reviewer for this comment. However, we double-checked this result, and it is correct.

Ref2#9. I was particularly interested by the results regarding the COVID-19 test results variable (first time reported to my knowledge) and especially by the results of the Unknown condition as it suggests that individuals think about risk in a “probabilistic” way (50% chance the individual is sick: medium distance). I was a little disappointed the authors did not develop more in the discussion section about those results. I think it is a key point of this research. 

ARRef2#9: We thank the Reviewer for pointing out this. We know expanded this point in the Discussion section (page 24-25, lines 354-368), which currently reads: “In particular, we found a continuous increase in the space that participants put between themselves and another person, with the shortest distance reported in association with a negative-tested individual, a medium distance observed when the other individual had an unknown-test result, and a maximum distance when the other person tested positive to COVID-19. These results may reflect the notion of “behavioral immune system” [54], according to which humans use behavioral avoidance of disease-causing objects and people as a disease-management strategy. The evidence that our three different conditions are associated with a continuously increasing space between participants and the another person suggests that the purported behavioral immune system may be regulated by a probabilistic inference about risk [55]: the higher the perceived risk, the larger the IPD. Indeed, when participants were not informed about the other person’s COVID-19 test result (i.e., Unknown condition) they might have relied on the conviction that the other had a 50% chance of being infected, thus placing themselves between the more extreme conditions (where a 0% risk is associated with the Negative condition and an estimated 100% risk to the Positive one).”

Reviewer #3: In a web-based experimental study conducted during the first pandemic wave (mid-April 2020), the authors tested preferred interpersonal distance with confederates. The variables explored concerned the role of protective equipment, actual risk of infection, perceived vulnerability, severity of the situation, physical and virtual contacts, morality, attitudes toward quarantine, and prosocial tendencies in the regulation of IPD during the COVID-19 outbreak. The test was based on the Interpersonal Visual Analogue Scale (IVAS), adapted to the present study. Based on Bayesian analysis approach the authors found evidence in favor of a reduction of interpersonal distance towards individuals wearing protective equipment and who tested negative to COVID- 19. Shorter interpersonal distances were also found with confederates wearing protective gear, even when COVID-19 test result was unknown or positive. Individual differences did not modulate significantly interpersonal distances. The protective equipment-related regulation of interpersonal distance may reflect an underestimation of perceived vulnerability to infection. Consequences in terms of collective health-safety measures communication are discussed. 

The aim of this study was to investigate whether being at risk of infection or having specific personal characteristics modulates IPD. The effects of variables were tested using a model selection approach embedded in a Bayesian analysis approach. The study was a web-based experimental study, but provided interesting results. I have only few comments relating to the state of the art in the research domain, and the analysis of the data, exposed hereafter.

We thank the Reviewer for his/her positive evaluation of our work.

Ref3#1. Introduction

When stating “handwashing and use of face masks have been widely adopted in conjunction with maintaining interpersonal distances of at least 1.5 m”, this is in fact dependent on the country (see for instance https://theprint.in/theprint-essential/1m-1-5m-2m-the-different-levels-of-social-distancing-countries-are-following-amid-covid/449425/)

ARRef3#1: We thank the Reviewer for this suggestion. We corrected this erroneous information in the abstract (page 2, lines 3-5): “The interpersonal distance of at least 1 m recommended as a relevant measure for COVID-19 contagion containment requires a significant change in everyday behavior.”

We also accordingly corrected this in the introduction (page 3, lines 29-36). The sentence now reads as follows: “These closures and other measures of transmission containment, such as handwashing and use of face masks [2], have been widely adopted in conjunction with maintaining interpersonal distances of at least 1 m [3]. The need to regulate the minimum distance during in-person interactions is justified by the observation that, although humans tend to keep themselves at about 1 m from unfamiliar individuals [4], this distance reduces when interacting with acquaintances and friends [5]. Crucially this pattern seems to hold across different countries [5], suggesting that the imposed governmental measures sought to change a globally established, everyday behavior”

Lastly, we corrected the information in the discussion (page 24, lines 344-345): “Interpersonal distance of at least 1 m is a fundamental measure of containment for the spreading of SARS-CoV-2.”

Ref3#2. When indicating “Research in proxemics, the study of interpersonal spatial behavior …” you should quote Hall (1966), who is at the origin of the research field.

ARRef3#2: We thank the Reviewer for this comment. We now cited Hall when defining interpersonal distance (page 3, lines 41-43): “Research in proxemics, the study of interpersonal spatial behavior [7,8], has defined interpersonal distance (IPD) as the separation zone that individuals keep between themselves and others [9].”

Ref3#3. When mentioning “IPD is shaped by situational factors such as social threat…” you should mention the demonstration made by Cartaud et al. (2018, Frontiers Psychology),

ARRef3#3: We thank the Reviewer for this suggestion. We now cited Cartaud et al., 2018 at page 3, line 43.

Ref3#4. When indicating “IPD appears to be automatically regulated according to distance-related feelings of personal comfort », you should quote and perhaps discuss the recent model proposed by Coello & Cartaud (2021, Frontiers in Human Neuroscience).

ARRef3#4: We thank the Reviewer for this suggestion. We believe that this reference better fits our Discussion section, when examining possible future directions such as the investigation of the neural basis of IPD regulation (page 27, lines 419-428, see ARRef3#13).

Ref3#5. Data analysis

It is not clear why sometimes the model includes one additional variable (Model 2 for instance) and sometimes two (Model 3 for instance).

ARRef3#5: We apologize for the lack of clarity on this point. We have now reported the corrected analysis, where only one variable was additionally included in each model. 

Ref3#6. Please, when discussing the Final Model, indicates that this refers to Model 4.

ARRef3#6: We thank the Reviewer for suggesting this. In the new analysis, Model 5 is the final one. We added the following sentence to page 14-15, lines 264-267: “Model 5 had the best predictive accuracy and included the same structure of the Primary model plus the main effects of Perceived Severity of the situation in the country, Virtual and Physical Contact, and Perceived Vulnerability to Disease (see Table 3).” For the sake of clarity, we also specified this at page 18, line 284: “Analysis of the final model (Model 5) focused on posterior contrasts between all levels of categorical predictors and the slope of continuous predictors.”

Ref3#7. When discussing Fig 2, please indicate what negative/positive directions correspond to in terms of the measure.

ARRef3#7: We now specified this (page 23, lines 325-335): 

“Fig.2 Probability of direction and the magnitude of the effect for each predictor included in the study.

The y-axis indicates the predictors and the x-axis indicates the possible parameter values. The color indicates the direction of the effect: black stands for a negative direction (reduction of IPD), while gray represents a positive direction (enlargement of IPD). The effect of the parameters included in the final model where HDI are completely outside of zero are marked with either “-” (if the direction of the effect is negative) or “+” (if the direction of the effect is positive). The interaction between COVID-19 Test Result and Protective Equipment and the interaction between Participant’s Gender and Other Avatar’s Gender are better explained by the contrasts between all levels of the factors (COVID-19 Test Result: Protective Equipment see Table 4 and Fig 3; Participant’s Gender: Other Avatar’s Gender see Table 5).

Ref3#8. I suggest to indicate which difference is significant in Figure 2. For instance, Gender effect (participants) is significant but this is not clear from the representation used in Figure 2.

ARRef3#8: We thank the Reviewer for this comment. We now report the non-zero effects in Figure 2 with a “-” symbol when the direction of the effect is negative and a “+” symbol when the direction is positive. See above (ARRef3#7) for the modified caption of Fig 2.

Ref3#9. You indicate that « IPD was shorter for women compared to men », but Figure 2 shows the opposite.

ARRef3#9: We apologize for this mistake. We now corrected it (pages 19, lines 307-309): “the preferred IPD was larger for women compared to men (estimate= 5.08, HDI [0.08, 10.06], BF10 = 0.02)”.

Ref3#10. Discussion

It can be indicated that people wearing a mask are also perceive as more trustworthy (Cartaud et al., 2020).

ARRef3#10: We thank the Reviewer for suggesting this. We now added the following sentence to our Discussion (page 25, lines 377-379): “Results showed that shorter IPD was judged as more appropriate for the characters wearing a mask compared to the other conditions; these characters were also perceived as more trustworthy.”

Ref3#11. When indicating “Among the other predictors, only Virtual Contact had a non-zero effect”, this is not in agreement from the information provided in the data analysis section where it is mentioned a “non-zero effects for Participant’s Gender and Virtual Contact”.

ARRef3#11: We apologize for not being clear on this. The sentence now reads as follows (page 26, line 390-391): “Among other predictors, Virtual Contact had a non-zero effect, meaning that frequent virtual contact led to larger IPD compared to infrequent virtual contact.” Moreover, we also discussed the effect of Gender in the following paragraph (see ARRef3#12).

Ref3#12. The Gender effect is not discussed and seems opposite to what is usually reported in the literature (for instance, Iachini et al., 2016, Journal of Environmental Psychology)

ARRef3#12: We thank the Reviewer for pointing this out. Regarding the main effect of the Participant’s Gender, what we found is in accordance with the relevant literature, as female (vs male) participants tend to show larger interpersonal distance when interacting with strangers (Iachini et al., 2014; Sorokowska et al., 2017). The opposite trend is generally found for the Confederate’s Gender, as participants of both genders tend to keep a shorter distance from female confederates compared to male ones (Iachini et al., 2016). In this new version of the MS, we now discuss the main effect of Participant’s Gender by taking into account the traditional literature as well as evidence regarding gender differences in the pandemic response. For example, evidence shows that men are less likely to believe that contracting the coronavirus may have serious consequences for them (Capraro & Barcelo 2020), and overall, they tend to comply less with preventive behaviors compared to women (Olcaysoy et al.,2020). Considering that recent evidence shows that the Gender effects are modulated by the sexual orientation of individuals (Welsch et al., 2020; Lisi et al., 2021) we also added a new model (Model 10) which includes a three-level interaction between Participant’s Gender, Other Avatar’s Gender and Participant’s Sexual Orientation (see ARRef2#5). This model, however, did not significantly improve the fit.

The new paragraph now reads as follows (page 26, lines 401-409): “Overall, women kept a larger distance from others compared to men, although Bayes factor analysis did not show decisive support for this effect. Women, indeed, tend to exhibit more defensive behavior during interactions with strangers [5,58]. Interestingly, this result, which requires further investigation, is in line with research on gender differences in the pandemic response, which showed that men’s belief of being gravely affected by COVID-19 is reduced with respect to women [59]. Additionally, men appear to be less likely to comply with preventive behaviors [60]. Contrary to previous evidence [12,14], participant’s sexual orientation did not modulate the gender differences, suggesting that, in the context of a pandemic, its relevance may be reduced.”

Ref3#13. When concluding that “the results of our study must be considered in light of recent neuroscientific evidence, which shows that the distance from other organisms (conspecific or not) is regulated by a common network … representing peripersonal space”. I am not sure about what means “common network” in this sentence. Furthermore, on this issue I recommend the authors to refer to the recent paper by Coello & Cartaud (2021, Frontiers in Human Neuroscience), who propose a new model to account for the relation between IPD and peripersonal space.

ARRef3#13: We thank the Reviewer for this suggestion, and we apologize for an inaccurate use of the term “common”. We have now referred to the perspective paper by Coello & Cartaud (2021, Frontiers in Human Neuroscience) and the paragraph reads as follows (page 27, lines 419-428): “Finally, the results of our study must be considered in light of recent neuroscientific evidence [61], which shows that IPD regulation may be rooted in the peripersonal space representation (the multisensory motor area within which it is possible to reach and interact with objects [62]). Indeed, Vieira and colleagues [63] showed that distance from other organisms (conspecifics or not) is regulated by a network that includes the midbrain periaqueductal gray (a defensive region, sensitive to threat proximity) and frontoparietal structures representing peripersonal space. Future neuroimaging studies may allow to investigate whether the reduction of IPD associated to seeing another person wearing a mask reflects a modulation of the activity in the above network, therefore supporting the hypothesis of a reduced perceived threat.”

---

## [Decision Letter · Decision Letter 1]

15 Jun 2021

PONE-D-21-03526R1

A Bayesian approach to reveal the key role of mask wearing in modulating projected interpersonal distance during the first COVID-19 outbreak

PLOS ONE

Dear Dr. Lisi,

Thank you for submitting your manuscript to PLOS ONE. After careful consideration, we feel that it has merit but does not fully meet PLOS ONE’s publication criteria as it currently stands. Therefore, we invite you to submit a revised version of the manuscript that addresses the points raised during the review process.

We look forward to receiving your revised manuscript.

Kind regards,

Valerio Capraro

Academic Editor

PLOS ONE

Journal Requirements:

Additional Editor Comments (if provided):

One of the reviewers suggests one final change before publication. Please address this last comment at your earliest convenience. Also, I would like to mention a recent review article on the role of prosociality in covid-19 response. Of course, it is not required to cite it, but, since you work on the same topic, I'm mentioning it because you may find it useful. I am looking forward for the final version.

Capraro V, Boggio PS, Böhm R, Perc M, Sjåstad H (forthcoming) Cooperation and acting for the greater good during the COVID-19 pandemic. In M. K. Miller (Ed.) The social science of the COVID-19 pandemic: A call to action for researchers. Oxford: Oxford University Press. Available at: https://psyarxiv.com/65xmg/

Reviewers' comments:

Reviewer's Responses to Questions

**Comments to the Author**

1. If the authors have adequately addressed your comments raised in a previous round of review and you feel that this manuscript is now acceptable for publication, you may indicate that here to bypass the “Comments to the Author” section, enter your conflict of interest statement in the “Confidential to Editor” section, and submit your "Accept" recommendation.

Reviewer #2: All comments have been addressed

2. Is the manuscript technically sound, and do the data support the conclusions?

Reviewer #2: (No Response)

3. Has the statistical analysis been performed appropriately and rigorously? 

Reviewer #2: (No Response)

4. Have the authors made all data underlying the findings in their manuscript fully available?

Reviewer #2: (No Response)

5. Is the manuscript presented in an intelligible fashion and written in standard English?

Reviewer #2: (No Response)

6. Review Comments to the Author

Reviewer #2: The authors have adequately addressed my comments. This manuscript is now acceptable for publication.

I however recommand modifying the periaqueducal grey definition "(a defensive region...)" which is, in my opinion, a bit awkward.

7. PLOS authors have the option to publish the peer review history of their article (what does this mean?). If published, this will include your full peer review and any attached files.

Reviewer #2: **Yes: **Alice Cartaud

---

## [Author Response · Author response to Decision Letter 1]

16 Jul 2021

Reviewer #2

The authors have adequately addressed my comments. This manuscript is now acceptable for publication.

I however recommand modifying the periaqueducal grey definition "(a defensive region...)" which is, in my opinion, a bit awkward.

ARRef2: 

We thank the Reviewer for pointing out this. 

We modified the sentence (page 27, lines 422-425), which now reads as follows: “Indeed, Vieira and colleagues [64] showed that distance from other organisms (conspecifics or not) is regulated by a network that includes the midbrain periaqueductal gray (a region sensitive to threat proximity and involved in defensive behaviors) and frontoparietal structures representing peripersonal space.”

---

## [Editor Report · Decision Letter 2]

21 Jul 2021

A Bayesian approach to reveal the key role of mask wearing in modulating projected interpersonal distance during the first COVID-19 outbreak

PONE-D-21-03526R2

Dear Dr. Lisi,

We’re pleased to inform you that your manuscript has been judged scientifically suitable for publication and will be formally accepted for publication once it meets all outstanding technical requirements.

Kind regards,

Valerio Capraro

Academic Editor

PLOS ONE
---

## [Editor Report · Acceptance letter]

29 Jul 2021

PONE-D-21-03526R2 

A Bayesian approach to reveal the key role of mask wearing in modulating projected interpersonal distance during the first COVID-19 outbreak 

Dear Dr. Lisi:

I'm pleased to inform you that your manuscript has been deemed suitable for publication in PLOS ONE. Congratulations! Your manuscript is now with our production department. 

Kind regards, 

on behalf of

Dr. Valerio Capraro 

Academic Editor

PLOS ONE